# Coupling between Osseointegration and Mechanotransduction to Maintain Foreign Body Equilibrium in the Long-Term: A Comprehensive Overview

**DOI:** 10.3390/jcm8020139

**Published:** 2019-01-25

**Authors:** Luis Amengual-Peñafiel, Manuel Brañes-Aroca, Francisco Marchesani-Carrasco, María Costanza Jara-Sepúlveda, Leopoldo Parada-Pozas, Ricardo Cartes-Velásquez

**Affiliations:** 1Dental Implantology Unit, Hospital Leonardo Guzmán, Antofagasta 1240835, Chile; 2Faculty of Sciences, Universidad de Chile, Santiago 7800003, Chile; branesmd.1@vtr.net; 3Clínica Marchesani, Concepción 4070566, Chile; francisco@marchesani.cl (F.M.-C.); mconstanzajara@gmail.com (M.C.J.-S.); 4Regenerative Medicine Center, Hospital Clínico de Viña del Mar, Viña del Mar 2520626, Chile; dr.polo@ejerciciosalud.cl; 5School of Dentistry, Universidad Andres Bello, Concepción 4300866, Chile; cartesvelasquez@gmail.com; 6Institute of Biomedical Sciences, Universidad Autónoma de Chile, Temuco 4810101, Chile

**Keywords:** oral implants, osseointegration, marginal bone loss, immunomodulation, mechanotransduction

## Abstract

The permanent interaction between bone tissue and the immune system shows us the complex biology of the tissue in which we insert oral implants. At the same time, new knowledge in relation to the interaction of materials and the host, reveals to us the true nature of osseointegration. So, to achieve clinical success or perhaps most importantly, to understand why we sometimes fail, the study of oral implantology should consider the following advice equally important: a correct clinical protocol, the study of the immunomodulatory capacity of the device and the osteoimmunobiology of the host. Although osseointegration may seem adequate from the clinical point of view, a deeper vision shows us that a Foreign Body Equilibrium could be susceptible to environmental conditions. This is why maintaining this cellular balance should become our therapeutic target and, more specifically, the understanding of the main cell involved, the macrophage. The advent of new information, the development of new implant surfaces and the introduction of new therapeutic proposals such as therapeutic mechanotransduction, will allow us to maintain a healthy host-implant relationship long-term.

## 1. Introduction

Titanium dental implants are inserted directly into the bone tissue, a complex and dynamic tissue. This bone tissue not only participates in calcium homeostasis and functions as a hematopoietic organ, but also plays an important role as a regulator of immunity [1].

Recent evidence on foreign body reactions (FBRs) in relation to implantable devices, such as titanium dental implants, reveals that, to achieve a lasting relationship between the implant and the host, titanium implants must have an optimal surface [2], and there must be an adequate healing capacity of the host [3]. Recently, it has been shown that the presence of a titanium implant during bone healing activates the immune system and displays type 2 inflammation, which seems to guide the relationship between the host and the implant [4]. This appears to indicate that osseointegration is a dynamic process, the result of a complex set of reactions in which several mechanisms and pathways of the host interact [1]. If the osseointegration is not altered, a continuous equilibrium occurs in the form of Foreign Body Equilibrium (FBE), which has been documented for 20 years or more in oral implantology [5]. Despite the high rates of survival achieved with titanium dental implants [6], it is necessary to further improve the implant-host relationship to maintain the integrity of the FBE long term; especially when the mechanisms involved in the breakdown of the osseointegration begin to act [7]. Once this occurs the immune system could be activated changing the delicate balance between the osteoblast and the osteoclast, which results in bone resorption [8].

The role of macrophages in osseointegration is greater than expected [1]. Macrophages respond to all implanted materials, which play an essential role in the fate of an implant [9]. Currently, immunomodulation strategies targeting macrophages are being developed around implants, both dental and orthopedic ones; either through new surface treatments [10], the controlled release of specific ions [11] or through specific cytokines [12]. The immunomodulatory effect of the Mesenchymal stem cell (MSC) has also been explored [13], and in this line, the hypothesis of immunomodulation of osseointegration through therapeutic mechanotransduction has recently been proposed, particularly by extracorporeal shock waves therapy (ESWT) [14].

The field of mechanobiology has allowed us to analyze the effects of mechanical forces on cellular processes [15], which has revealed the complex cellular regulation involved in the transduction of mechanical signals [16]. Mechanical stimuli can stimulate the activity not only of bone cells but also MSC [17]. Mechanical stimuli can also change the cellular form and affect the phenotype and function of immune cells, such as macrophage and dendritic cells [18].

This review begins with (i) a discussion of key concepts related to bone tissue and the immune system; (ii) next, we will discuss the FBR, focusing specifically on osseointegration; (iii) to then explore the current strategies of immunomodulation in osseointegrated implants (iv) Finally, we will conclude with a discussion on a topic that may become clinically relevant, the coupling between osseointegration and mechanotransduction to maintain FBE long-term.

## 2. Bone Tissue and Immune System

The scientific field of osteoimmunology has revealed the vital role of immune cells in the regulation of bone dynamics [19], this has led to the understanding of the existence of different molecular and cellular mechanisms involved in a permanent interaction between bone tissue and the immune system. For this reason, to understand bone healing in general and osseointegration in particular, it is necessary to understand the biology and immunology of bones [20]. Bone is an organ composed of cortical, trabecular, cartilaginous, hematopoietic and connective tissue [21], which is composed by more than 30 different cell populations which reside in the microenvironment of the bone marrow adjacent to an implant. These cell populations, alone or in combination, have the ability to influence the formation and the bone regeneration of the peri-implant environment [2]. In addition, the presence of multiple anatomical and vascular contacts allow for a permanent interaction between the bone tissue and the immune system [22]. In fact, the bone marrow shows structural and functional characteristics that resemble a secondary lymphoid organ. That is why bone marrow is currently considered an immunoregulatory organ, capable of significantly influencing systemic immunity [21].

The cells of the bone tissue and the cells of the immune system share common origins. Osteoclasts (OC) come from stem cells of the monocyte-macrophage cell lineage [22]. However, certain subclasses of circulating monocytes and dendritic cells (DCs) which reside in the bone marrow also have the capacity to transform into OC if they are subjected to certain specific signals [23]. Perhaps this common origin with cells of the immune system could be related to the ability of OCs to recruit CD8 + FoxP3 + T cells and present antigens to them [24,25]. On the other hand, osteoblast (OBs) play a central role in the differentiation of hematopoietic cells [22]. This common origin between osseous and immune cells facilitates understanding of how molecular pathways are involved in bone remodeling (such as in PTH, BMP and Wnt pathway) which also act in regulating the hematopoiesis [26]. 

Immune cells regulate osteoclastogenesis by three main cytokines: macrophage colony stimulating factor (M-CSF), receptor activator NF kappaB ligand (RANKL) and osteoprotegerin (OPG) [19]. The main element in osteoclastogenesis, the RANKL, can be expressed by activated T lymphocytes, dendritic cells and neutrophils, indicating the participation of these immune cells during osteoclastogenesis [19,21]. The expression of RANKL by activated T cells has been implicated in osteoclastogenesis induced by inflammation, linking adaptive immunity to skeletal biology [27]. This is related to the role of the immune system in several bone diseases, such as osteoporosis, osteoarthritis and rheumatoid arthritis. Several studies have clearly highlighted the role of developing T lymphocytes and the pathophysiology of osteoporosis, which has given birth to a new field of biology called “*immunoporosis*” [28]. Moreover, as the dendritic cells are responsible for the activation of virgin T cells and act as osteoclast precursors, this could be directly involved in osteoclastogenesis induced by inflammation and bone loss [29]. It has been described that persistent inflammation is characterized by the continuous release of proinflammatory cytokines (TNF-a, IL-1a/be IL-6), which is accompanied by a higher RANKL/OPG ratio and an increased osteoclast activity [30]. On the other hand, it has been shown that B cells are an important source of OPG derived from bone marrow, which implies that B cells are one of the main inhibitors of osteoclastogenesis in normal physiology [19].

Macrophages are precursors of osteoclasts, and under the stimulation of M-CSF and RANKL, they can differentiate into osteoclasts during bone remodeling [19]. Bone and bone marrow contain multiple subpopulations of specialized resident macrophages (bone macrophages or osteomacs), which contribute to bone biology and/or hematopoiesis [31]. Macrophages promote osteoblastogenesis in in vitro matrix deposition and they could have an important role in the promotion of bone anabolism, through the provision of trophic support to the osteoblast lineage [32]. In fact, the depletion of macrophages leads to the complete loss of bone formation mediated by osteoblasts in vivo [33]. In addition, they would be important in the reversion phase of a basic multicellular unit (BMU), which separates bone resorption and bone formation [34]. Macrophages are also abundant within the bone callus during the inflammatory phase of bone healing in humans [35], so the presence and diverse functionality of macrophages could allow an important contribution in bone homeostasis and throughout the course of bone healing [32]. Furthermore, the healing of bone fractures is significantly improved in knockout mice lacking T and B cells, which indicates that they may also have a detrimental function during this process. This observation suggests the dual role of immune cells in osteogenesis, through its expression and secretion of a wide range of regulatory molecules [19].

As we have seen, immune cells play an important role in bone homeostasis. Therefore, the insertion of a foreign body into the bone tissue will inevitably be recognized by the immune system, affecting the biological behavior of the bone cells. This event can determine the in vivo destination of an implant or “biomaterial”. 

## 3. Foreign Body Reaction and Osseointegration

The interaction between bone tissue and implants involves at least 3 components: immune cells of the host, bone cells of the host and the material [19]. After implantation, the host will experience a response to tissue injury, which will be conditioned by the material present and the degree of the immune response [36].

In general, after a surgical implantation procedure, the damage of the endothelial cells exposes the underlying vascular basement membrane and initiates the coagulation cascade that leads to the formation of a clot of red blood cells-platelets-fibrin. This vascular damage also facilitates the interaction of the implant with blood proteins and interstitial fluids, such as fibrinogen, vitronectin, complement, fibronectin and albumin, which are adsorbed dynamically on the surface of the implant (Vroman effect) in seconds, forming a superficial transient matrix [1,2,7,19,36,37]. This allows physicochemical interactions between the host proteins and the implant’s surface, which leads to a change in the molecular conformation of one or more of these host proteins, exposing previously hidden amino acid sequences, which would act as antigenic epitopes [1,7]. Serum factors called “opsonins” will participate in the recognition of the foreign agent, the main ones being immunoglobulin G (IgG) and the complement activated fragment, C3b, allowing for interactions with macrophages through membrane receptors [38]. Hu et al. showed that adsorbed fibrinogen is the main protein responsible for the accumulation of macrophages on the surfaces of implanted biomaterials [39]. It has also been shown that adsorbed fibrinogen exposes two previously hidden amino acids, functioning as epitopes, which allows for interaction with macrophages through the Mac-1 integrin (CD11b/CD18), leading to a proinflammatory environment, modulating the response of the host to the biomaterial in this way [1]. In this same context, it has been suggested that another protein, fibronectin, could participate in the chronic phase of FBR [40].

The complement system seems to play a key role at this early stage [7]. Arvidsson et al. showed that the interaction between titanium and plasma coagulation factors, such as factor XII, could lead to the activation of the complement through the alternative pathway, producing C3b [41]. As is known, immune cells express inactivated C3b/C3b (iC3b) receptors, so that from the early phase of inflammation, the surface of the implant is recognized by the immune system [7]. Recently it has been demonstrated in titanium implants that there is a positive regulation of the C5a-1 receptor (C5aR1) after the inflammatory period, which demonstrates a prolonged activation of innate immunity through the continuous activation of the complement system [4].

After the initial interaction between the blood and the material, acute inflammation begins, which is initiated by the cytokines and chemokines released by the damaged cells, leading to the influx of neutrophils and mononuclear macrophages [19,36]. Neutrophils normally deplete rapidly, undergo apoptosis and disappear from implantation sites within the first two days [19]. The prolonged presence of neutrophils indicates that we are facing active chronic inflammation [36]. This has been observed around titanium implants, probably due to the role of neutrophils in the promotion of vascularization in tissue hypoxia and the ability of the macrophage to suppress the apoptosis of them [4]. It is important to mention that neutrophils, in an effort to degrade the materials, release proteolytic enzymes and reactive oxygen species (ROS), which can corrode the surface of the implanted material [19].

The influx of mononuclear macrophages occurs between 24 and 48 hours, which have a phagocytic function that includes the release of proteolytic enzymes that degrade cellular debris and the extracellular matrix (ECM). Currently, these immune cells have aroused great interest among scientists due to their multiple functions in the process of bone healing and high plasticity [19]. Macrophages have been extensively characterized in phenotypes M1 and M2, reflecting the Th1/Th2 nomenclature described for helper T cells [42]. Traditionally, it has been described that M1 proinflammatory macrophages would dominate the early phase of the reparative response and, on the other hand, M2 macrophages (M2a, M2b and M2c), would play a more prominent role during the middle and later stages of the response repair [19]. However, this classification represents only a simplification of the in vivo scenario, since it is very likely that the macrophage phenotype occupies a continuum between the M1 and M2 designations, with transient macrophages with characteristics of both phenotypes present [43]. Therefore, it seems that both phenotypes of macrophages perform essential functions during the process of bone healing, with the macrophage change pattern determining the osteogenesis instead of a specific macrophage phenotype [44].

It has been described that a prolonged M1 polarization phase leads to an increase in fibrosis-enhancing cytokine release pattern by the M2 macrophages, which results in the formation of a fibrocapsule around the biomaterial [19]. This reaction could be related to the “primary failure”, which occurs in 1–2% of all dental implants placed, probably due to a series of risk factors that are predisposed to this total failure in osseointegration, like the following: low primary stability, premature loading, traumatic surgery, infection, as well as patient conditions such as smoking or the consumption of some pharmaceutical products [20].

On the contrary, an efficient and timely switch from M1 to M2 macrophage phenotype results in an osteogenic cytokines release and with it the formation of new bone tissue [19]. This second possible reaction is one that would generate the bone encapsulation that allows the commercial use of titanium implants [20]. In commercially pure (c.p.) titanium implants, the presence of the M2 phenotype of the macrophage (most likely M2a) is significantly high, which has been observed as early as 10 days after surgery [4,45]. This indicates that there is an immunomodulated relationship between the titanium implant and the host, which allows the deposit of bone in the implant, and the isolation of this from the space of the bone marrow, through a type of FBR [4]. Albrektsson and colleagues introduce the concept of FBE to describe this phenomenon, osseointegration being considered a mild chronic inflammatory response that allows implant function with a bone-implant interface that remains in a state of equilibrium, susceptible to changes in the environment [5].

Macrophages can swallow particles up to 5 μm, however, if the size of the material or the residue is greater than 50–100 μm, the material is surrounded by macrophages that fuse to form foreign body giant cells (FBGC) [19,46]. This induction of macrophage fusion probably occurs through the secretion of IL-4 and IL-13 by mast cells, basophils, and helper T cells (Th) [40]. It has even been suggested that FBGCs could express phenotypes of M1 and M2 macrophages, depending on the environment, similar to their mononuclear precursors [47]. Although FBGCs are not normally found in healthy tissues, they are abundant around implanted biomaterials, even years after implantation [48]. This is the reason why the presence of FBGC in the interface of the host-implant is an indication of an FBR to the implanted material or device [19,40,46]. Donath et al. described the presence of FGBC on the surface of titanium implants, which were present in multiple cases of FBR through histological studies [49]. It has been seen that the CD11b marker is extremely upregulated at 28 days, demonstrating how macrophages are highly involved in the reaction to titanium implants since this marker has recently been implicated in the fusion of macrophages [4]. For several years, multinucleated giant cells (MNGC) have been described in relation to biomaterials, especially in the case of bone replacement materials, assuming that MNGCs are osteoclasts. However, many studies indicate that these cells actually belong to the cell line of FBGCs, which are of an "inflammatory origin" [47]. It is a fact that osteoclasts can be formed by the fusion of multiple macrophages, and some authors even suggest that macrophages can perform functions of bone resorption [50]. All of the above suggests that FBGS could play a central role in the pathway of bone loss during the FBR [7].

Macrophages and dendritic cells can initiate an adaptive immune response through the presentation of antigens, which can also be particles or ions. When a T cell recognizes an antigen, the T cell is activated (activated TCD4 +) and may have inflammatory secretory profiles (Th1) or anti-inflammatory secretory profiles (Th2) [51]. Recently a constant regulation of CD4 and the negative regulation of CD8, which indicates a reaction of CD4 lymphocytes around the implant, was observed in titanium implants [45]. However, more research is needed to confirm the continued presence of the immune system over time [4].

As we have seen, an implanted device activates the components of the immune system in bone: complement, neutrophils, macrophages and lymphocytes. However, the role of the macrophage in the host-biomaterial relationship is highlighted [4]. Although most implants will be successful, a rejection mechanism may occur. This may be represented by marginal bone loss around the osseointegrated implant, which could be a product of multiple factors such as the implant, the surgical procedure, prosthetic conditions and factors in relation to the patient [20]. The loss of FBE could be the main cause of this peri-implant bone loss. This leaves the door open for the development of different strategies to face the pathology through a deeper understanding of the biology of osseointegration [1].

## 4. Current Immunomodulation Strategies in Osseointegrated Implants

Titanium is one of the few materials suitable for implantation requirements in the human body, being widely used in oral surgery, maxillofacial surgery, craniofacial surgery and orthopedics. The greater clinical use and popularity of oral implants have led to a growth in demands, with an increasing need for treatments in places where the quality of bone is less than ideal, and in patients with a compromise of scarring products of systemic affections [2]. Although less than 5% of oral implants show failure under optimal clinical conditions. In some cases, through a triad consisting of poor clinical handling, combined with poor implant systems and the treatment of those who are compromised, treating “poor” patients may lead to problems, probably increasing the number of complications [20]. On the other hand, many orthopedic procedures require implants, however, not all implanted devices last forever: up to 15% of the total joint implants require a surgical revision within 15 years of the initial surgery [52]. Therefore, there is a need to improve the biological function and longevity of the implantable devices [20,52].

Recent studies on osseointegrated titanium oral implants demonstrate the presence of the M2 phenotype of the macrophage from the early stage of healing (days) [4,10,45,53]. However, the presence of other chemical elements on the surface of the titanium implant seems to be relevant for the bone balance of osseointegration [2]. Trindade et al. have demonstrated that the bone resorption markers were significantly down-regulated around titanium in turned titanium grade IV implants. Interestingly, the regulation balance of bone resorption RANKL/OPG is suppressed in its entirety, suggesting that bone resorption has been kept to a minimum around Titanium [4]. However, Biguetti et al., using a machined titanium implant of titanium-6 aluminum-4 vanadium alloy, demonstrated the opposite; there was a remarkable remodeling process, evidenced by peaks corresponding to RANKL and OPG, and also an increased area density of osteoclasts. Furthermore, the presence of ten chemical elements in the surface composition of the implants used was determined through an analysis by Energy Dispersive X-ray (EDX): Titanium [Ti], Aluminum [Al], Vanadium [V], Calcium [Ca], Nitrogen [N], Niobium [Nb], Oxygen [O], Phosphorus [P], Sulfur [S] and Zinc [Zn] [53].

In this context, it is noteworthy that c.p. titanium is often alloyed with aluminum and vanadium (Ti6Al4V). However, in some cases, further surface modification procedures such as sand-blasting and acid etching are likely to remove passive layers from the surface of the metal, exposing less stable elements underneath. This could generate an inflammatory response and possible reduction in osteoblast differentiation [2]. Both of these effects can be detrimental to new bone formation and implant integration. However, in relation to the aforementioned, this material has an acceptable clinical success currently [10]. However, the degree of purity of the surfaces is an important issue to consider since there are important studies of oral implants, which reveal the presence of organic and inorganic contaminants onto some surfaces [54,55].

The above could be clinically relevant since, as we know, macrophages respond to all the implanted materials, being fundamental for the fate of an implant [36]. Macrophages are capable of releasing metal ions from solid surfaces in a matter of minutes by dissolucytosis [2]. The fused macrophages in FBGC can remain in the interface biomaterial-tissue, generating a sealed compartment between its surface and the underlying biomaterial, which allows the secretion of different mediators such as ROS, degradative enzymes and acid. Due to this the "frustrated phagocytosis process” being associated with the failure of some implanted devices [48]. Particles, ions, or degradation products from implanted materials or devices may also be recognized as foreign by macrophages and dendritic cells [9,56]. Dendritic cells may also be drawn to the implant site by the recognition of foreign substances, inducing the expansion of CD4 cells [9,21], so that some dental implant could eventually be able to cause a type IV hypersensitivity reaction [57]. Since the bone marrow contains structures in the form of follicles, similar to that observed in lymph nodes or the spleen, although without an organized T and B zone, but these lymphoid follicles can increase in the bone marrow during infections, inflammations and autoimmunity [21].

It has been demonstrated that titanium leakage due to corrosion inevitably results in substantial contact between the foreign material and the tissues. In fact, there was a gradient in titanium intensity from the implant surface and out up to a distance away from it of about 1000 μm [20]. Titanium ions could cause immune responses due to their ability to bind to proteins, such as albumin or transferrin, creating a bioavailable metalloprotein that could serve as an antigen in immunological reactions. Many studies have shown that proinflammatory cytokines such as IL-1β (interleukin 1beta), TNF-α tumor necrosis alpha factor, and GM-CSF (granulocyte-macrophage colony stimulating factor) are jointly regulated after stimulation of a hapten or particles [58]. However, it is likely that this first corrosion is coupled to the acidic environment that inevitably develops after the placement of an implant, a situation that is present until approximately four weeks after surgery when the partial pressure of oxygen has normalized [20]. In all probability at this stage, the released ionic titanium is stabilized by biomolecules such as citrate, an important metal chelator in cellular fluids, forming relatively stable complexes in solutions close to neutral pH [58]. Thereafter it seems likely that titanium corrosion will be quite minimal, provided that there is no more mechanical interruption of the blood flow. The presence of titanium ions in a stage subsequent to osseointegration could generate a synergistic interaction with other negative factors, such as cement particles, leading to marginal bone loss [20].

This scenario where living tissues face the presence of materials in an immunologically active environment allows for a better understanding of the dynamics of osseointegration, and also reveals that the desired FBE in an oral implant can be threatened by clinical conditions [7]. This is why the methods that control the polarization of the macrophage have emerged as an attractive means to reduce inflammatory signaling [19]. It is known that the increase in bone formation correlates with the resolution of the initial inflammatory response. That is, inflammation and osseointegration are inversely proportional [2]. As osseointegration is an immunomodulated inflammatory process, where the immune system is locally up-or down-regulated [57], the precise modulation of postoperative inflammation and the innate immune reaction provide a promising approach for therapeutic purposes [19]. 

Several immunomodulation strategies have been proposed, mainly through the implanted device, and more recently, the immunomodulation by means of direct stimulation of Human bone marrow-derived mesenchymal stem cell (HBMMSC). This aims to improve integration, avoid fibrosis, prevent bone loss and so increase the useful life of the devices in the human body [14,19,59].

In relation to these immunomodulation strategies through implanted devices, the topographic modification of the titanium dental implant surface has shown a significant positive effect in the speed and degree of osseointegration. In fact, the use of microscale modified implant surfaces has been one of the key factors in increasing the clinical success rate of implants, especially in areas of compromised bone quality [2]. The surface topography of the implant can be optimized at a micro level and nanoscale, influencing properties such as wettability and surface charge, modifying the kinetics of adsorption, the folding of proteins adsorbed onto implant surface and the consequent presentation of bioactive sites to macrophages [51,60]. The geometry of the material may also be relevant for the phenotypic expression of macrophage. It has been demonstrated that micro- and nanopatterned grooves of 400–500 nm wide can influence macrophage elongation, driving macrophages toward an anti-inflammatory, pro-healing phenotype [61]. This interaction of the macrophage with its mechanical environment, that is, the surface of the titanium, is possible through multiple mechanoreceptors on the cell surface, perhaps through integrins [62]. At present, titanium can be alloyed with Zirconium (TiZr). The combination of high-energy and altered surface chemistry (hydrophilic), seems to generate an immunomodulatory effect towards the activation of M2 macrophages, decreasing the presence of FGBC and increasing osseointegration [10].

A growing number of studies report success based on therapies with metal ions such as magnesium, strontium, calcium, among others. These ions can be incorporated into devices, in order to promote osteogenesis coupled with a pro-regenerative immune response. For example, the production of proinflammatory cytokines such as TNF-α, IL-1b, IL-6 and PEG2 has been shown to be reduced in the presence of high concentrations of magnesium, which highlights its role as an immunomodulatory ion [63].

The incorporation of immunomodulatory molecules to the implant constitutes another strategy to modulate the immune response [12,52]. Inflammation could be controlled by the local release of M2 polarizing cytokines such as IL-4, IL-10, IL-13 [64], or the implant could directly inhibit proinflammatory signals, using anti-TNF-α therapy, which are the most potent proinflammatory cytokines that promote the polarization of M1 macrophages [65]. The transcription factor NF-κB has also been pointed out as a possible target to generate implant-mediated immunomodulation. It has been shown that NF-κB decoy can suppress the production of essential chemokines for the recruitment of monocytes, which could avoid the presence of immune cells at the bone-implant interface [66].

In spite of the above-mentioned issues, the biology is more complex. The determination of the appropriate time frame of immunomodulation is critical for optimizing their application. Acute phase inflammation is crucial for proper bone repair after trauma, so the macrophage polarization status also plays a critical role in bone regeneration. As such, the interplay between M1- and M2-dominated microenvironments and the temporal modulation of the transition M1 to M2 provide an interesting line of investigation to pursue new immune-modulatory therapies and improve bone repair and implant integration [52]. One possible method is to utilize a controlled release system to maintain a short period of M1, followed by a transition to M2 polarization via cytokines in a biphasic manner. However, future investigations are necessary [67].

A new therapeutic approach to achieve modulation of the transition from M1 to M2 in the appropriate timeframe could be the MSC of the host, given their innate immunomodulatory capabilities [68]. More than 400 studies have explored the immunomodulatory effect of MSCs for the treatment of various autoimmune conditions, including graft-versus-host disease, diabetes, multiple sclerosis, Crohn’s disease, and organ transplantation [69]. In relation to this, a new hypothesis has recently been proposed, that the HBMMSC residing around the peri-implant bone tissue immunomodulate the osseointegration process through the ESWT bio-activation effect. The mechanical stimuli generated by ESWT trigger the release of exosomes by HBMMSCs, generating tolerogenic dendritic cells (Tol-Dcs) and increasing the presence of the M2 phenotype of the macrophage [14].

## 5. Coupling between Osseointegration and Mechanotransduction to Maintain FBE Long-Term

At present we know that osteogenesis does not depend only on the bone cells of the skeletal system, in fact, there is a multicellular collaboration. Over years, studies have focused on the interaction between bone cells and the material surface, however, now studies in the field of advanced bone materials should involve co-culture systems with the interaction between materials, bone cells, and immune cells. Only then will we know the real osteoimmunomodulatory capacity of the material [19]. However, the complexity becomes even greater when we consider a fourth factor that can become important to maintain the balance of the material-host relationship, the mechanical stimuli. Physical forces also play important roles in embryonic development, tissue homeostasis, and pathogenesis. However, the importance of mechanical signals to control cellular processes has only been recognized more recently. That is why, as interest grows in the field of mechanobiology, new study models are developed to analyze the influence of mechanical forces on cells and tissues [70,71].

The cells are sensitive to shear, tension and compression forces. These mechanical signals have important effects on tissues, such as the production of ECM components [72]. A mechanical alteration can influence gene expression and cellular behavior through the mechanotransduction signal [73]. These mechanical signals would be transmitted by the filaments of the cytoskeleton, such as actin and microtubules, and finally transduced into biochemical signals [74], being the integrins of the cell surface essential for mechanotransduction [75].

The therapeutic mechanotransduction is part of modern Implantology, in fact, good clinical results obtained through progressive loading protocol [76] and immediate loading protocol [77,78], reveal to us that physiological mechanical stimuli can be beneficial to accompany the osseointegration of a dental implant and allow for successful osseointegration [79]. In this sense Duyck et al. demonstrated through a bone chamber model in an animal model that mechanical stimuli are capable of increasing the bone-implant contact (BIC) [80]. However, we also know that mechanical stimuli are key in bone remodeling, so greater trauma can lead to implant failure [20].

At present, ESWT are widely used in the context of therapeutic mechanotransduction. ESWT are supersonic waves, generated by different types of devices, such as electrohydraulic, piezoelectric, electromechanical or pneumatic, which generate transient pressure changes that propagate through the tissues where they are applied [81,82]. ESWT is applied to treat various medical pathologies. In orthopedics, it is used mainly in the treatment of tendinopathies, the treatment of nonunion in fractures of long bones, avascular necrosis of the femoral head, chronic diabetics, nondiabetic ulcers and ischemic heart disease [83]. In dentistry, ESWT has been used in extracorporeal lithotripsy of salivary stones [84] and painful mielogelosis of the masseter [85]. Recently, Falkensammer et al. used ESWT as a supplement in Orthodontics [86], finding an absence of deleterious effects in the maxillofacial tissues or for pulpal vitality [87]. 

It has been proposed that the mechanical stimuli generated by ESWT produce an increase in the permeability of the cell membrane, which triggers the release of cytoplasmic ribonucleic acid (RNA) through an active process dependent on exosomes. This event at the cellular level is the one that would produce the effects observed in the accelerated repair of tissues. However, more studies are needed to completely reveal the underlying mechanism [88].

It seems that the bone is programmed to seal immediately any area affecting its integrity, sealing and protecting the marrow content through the restoration of a cortical bone barrier. Therefore, we can assume that the "raw materials" (phosphate, calcium, etc.) could be more available from a source of cortical bone [4]. In this sense, it has been described that oral implants installed in low-density bone tissue (bone type IV) present a higher risk of failure [89]. On the other hand, in patients with osteoporosis, even though there is no difference described to date in the survival rate of oral implants placed in patients with and without osteoporosis, there is an increase in peri-implant bone loss [90]. In orthopedics, it is a proven fact that the fixation of screws and osteosynthesis plates can be hindered in patients with low bone mass and especially with thin cortices. In fact, in osteoporotic fractures, depending on the location, the type of fracture and the surgery performed, the failure rate can reach up to 30% [91]. In this sense, the anabolic effect on the bone described in several studies with the use of ESWT has become particularly attractive [82,92].

Recent research shows that this anabolic bone response through ESWT can also be generated in relation to titanium devices in bone, which could have great therapeutic potential, especially in patients with bone disease. Koolen et al. demonstrated at the histological level that in bone defects reconstructed with a titanium scaffold as a bone substitute show the de novo bone formation after ESWT in rats [93]. In this same line of investigation, Koolen et al. [91] hypothesized that peri-operative shock wave treatment can improve screw fixation and the osteointegration of cortical and cancellous orthopedic screws, especially in osteoporotic patients. They were able to demonstrate in a healthy rodent bones model, that an ESWT immediately after the implantation of titanium screws (Ti6Al4V grade 5) improved screw fixation of the cortical screw, visualized by improved mechanical strength and osseointegration. Another finding was the formation of a neocortex in some animals after treatment. However, the cancellous screw showed no differences in testing after ESWT [91]. In this context, it has been reported that the ESWT not only achieves complete bone healing, but has also been observed to help in the re-attachment of a loose orthopedic screw. This has been observed in a patient with a typical case of non-union treated with ESWT [94].

It has been suggested that these anabolic effects in the bone are due to the fact that shock waves can cause the conversion of progenitor cells into osteogenic precursor cells. Another possibility could be that ESWT induces osteocytic cell death through a mechanism called cavitation. This death of osteocytes could lead to the stimulation of local bone remodeling, activating the osteoblast to produce more osteoid, which could eventually lead to a neocortex [91]. I Osteocytes are important regulators of cellular homeostasis and can act as mechanosensors. In addition, direct contact between dendrites of the osteocytes and the implant surface has been reported after an 8-week osseointegration period in an in vivo model [1]. However, the presence of HBMMSC and immune cells in the peri-implant environment leads us to think that the ESWT could also have a potent immunomodulatory effect in favor of osseointegration [14].

HBMMSC are not only found in the peri-implant environment, but also adhere to the titanium surface [95]. Current evidence indicates that BMMSCs can modulate the immune response by inhibiting polarization induced to M1 macrophages and promote polarization to M2 macrophages through the release of paracrine factors [96]. In addition, HBMMSCs can modulate the immune response through the generation of Tol-DC [97]. This is why the fact that ESWT can act as an effective bioactivator on HBMMSC, increasing its rate of growth, proliferation, migration and reducing apoptosis of these cells, suggests that ESWT could be an adequate tool to express all the potential therapeutic effects of HBMMSC [98]. This evidence suggests that the findings described in relation to ESWT and titanium [91,93,94] are probably a product of the local immunomodulatory effect of HBMMSC [14].

Mechanical signals play an important role in the regulation of immunological and cellular processes in monocytes, macrophages, and dendritic cells [70]. In this sense, recent studies have investigated the anti-inflammatory effect of ESWT in ischemic lesions in the animal model. Scientists have observed that ESWT would regulate the inflammatory reactions reducing the infiltration of inflammatory cells and promoting the differentiation of the M2 macrophage, that is, through the immunomodulation effect [99]. Apparently, the ESWT also exerts direct modulation on the macrophage. It has been demonstrated that the stimulation of macrophages derived from human monocytes with ESWT causes the significant inhibition of some M1 marker genes (CD80, COX2, CCL5) in M1 macrophages and a significant synergistic effect for some M2 marker genes (ALOX15, MRC1, CCL18) in the M2 macrophages. It has also been observed that ESWT affected the production of cytokines and chemokines, inducing, in particular, a significant increase in IL-10 and a reduction in the production of IL-1β [100].

No doubt, infiltrating immune cells play an important role in determining the variable outcome of wound repair in mammals and amphibians. For example, it has been demonstrated that the systemic depletion of macrophages results in a permanent failure in the regenerative capacity of the axolotl, with extensive fibrosis. However, the regenerative capacity of the axolotl is recovered once the macrophage population is restored [101].

While the use of immunomodulatory implants per se (clean implants) generates adequate osseointegration [55], the FBE can be altered under certain clinical conditions, such as overload [1,5,7]. The possibility that we can guide the transition between the M1 inflammatory phase and the M2 anti-inflammatory phase through mechanotransduction makes ESWT a promising therapeutic alternative to improve clinical success in oral implants, maintaining FBE long-term [14]. This could potentially improve the feedback path to the sensory cortex since, as described, the capacity for tactile perception of osseointegrated implants, "osseoperception", increases over time [102]. Furthermore, it has also been proposed that the topical addition of the nerve growth factor (NGF) in oral implants could help to improve this tactile sensitivity in order to minimize occlusal overload [103], and it has been demonstrated that ESWT is effective in increasing the expression of NGF [104].

## 6. Concluding Remarks

Mechanotransduction can improve the implant-host relationship. However, it is necessary to perform studies at the cellular and molecular level that would allow us to determine both the medical device and the most effective therapeutic range. All of this is in order to improve, maintain and recover the harmony of this triad of elements, i.e., bone cells, immune cells and implants, which finally determines the fate of FBE.

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
