# Peer review of "Coupling between Osseointegration and Mechanotransduction to Maintain Foreign Body Equilibrium in the Long-Term: A Comprehensive Overview"

_jcm, 2019, doi:10.3390/jcm8020139_

Reviewer 1 Report

This is an interesting study, dealing with the clinically relevant question. Still, some concerns appeared during this review.

 “Comprehensive” in the Title would be more appropriate than “complete”.

Stile of writing should be improved. 

Please exclude personal impressions like “We believe” (page 1, line 23; page 10, line 486) or “we know” (page 2, line 84).

Some sentences are too long (page 1, line 43 – page 2, line 48; page 3, lines 121 - 125) or should be rephrased (page 2, lines 89 – 92; page 7, lines 328 – 330). There are also some typos (page 6, line 270).

The abbreviation ESWT should be included first time when “Extracorporeal shock waves” was mentioned in the text (page 8, line 365/395).

The importance of HBMMSC should be quoted earlier in the section 4.

Aseptic loosening is not a frequent complication in implant dentistry.

Peri-implantitis is not a secondary infection (page 6, line 249). Clinical symptoms of peri-implantitis are not a consequence of marginal bone loss; in clinical settings it is the opposite (page 5, line 234 – 236).

It seems necessary to repeat some statements, but it should be kept at minimum. The whole manuscript should be more concise and fluent, and thus easier to follow for the readership. 

Author Response

Dear doctor, I appreciate your comments and will make all the changes you have indicated to me as soon as posible, so you can review it.

with respect to peri-implantitis, you're right. I have not clearly explained this probable relationship between the loss of FBE and peri-implantitis. I think it is much better not to include those paragraphs in the paper, to keep the text focused on the mechanotransduction to maintain Foreign Body Equilibrium. But I would like to tell you my point of view about peri-implantitis: although implant surfaces that show problems leading to bone resorption have titanium surfaces with biofilms, a part of the biological puzzle is still not clear in relation to the progesion of this pathology. In fact, "The histopathological and clinical conditions that lead to the conversion of peri-implant mucositis to peri-implantitis are not fully understood" (Frank Schwarz Jan Derks Alberto Monje Hom-Lay Wang, Peri-implantitis, WORLDHOPKHOP 2017). That is why I think it is very important to maintain the FBE, since this biological balance could prevent the failure of defense mechanisms and the loss of marginal bone induced, for example, by bacteria. However, I believe that this marginal bone loss could occur aseptically, especially if proper osseointegration is not achieved, for example, due to the use of a poor quality and / or contaminated surface. This also could favor an secundary bacterial infiltration, product of a long time environment of Foreing Body Reaction, that is, an environment of many molecules, factors, mediators and signaling pathways. Perhaps this leads to a biological event similar to mesenchymal epithelial transition (EMT). The eventual presence  of antigens such as metal ions, in the implant surface, customised pillar abutments made of  chrome-cobalt or the presence of cement particles, can also stimulate dendritic cells and produce signs of soft tissue inflammation.

Because the molecular landscape involved in the progression of peri-implantitis is still not fully understood, just as you recommend, it is better to correct the text in this point.

Kind Regards

Luis Amengual-Peñafiel

Reviewer 2 Report

this is an excellent manuscript with novel ideas integrating osteointegration with immune mechanisms. the manuscript is well referenced and well  thought out.

A few questions which selected readers may ask are

what impact do the PMN Neutriphil enzymes form lysosomes have after apoptosis? Do the proteolytic and hydrolytic enzymes result in further bone resorption?

and

Macrophage genotype impact, if known?

Author Response

Thank you very much for your comments. The relationship of the implanted devices with the host is a topic of clinical importance that should be further investigated, especially at the immunological level.

Point 1: what impact of the PMN Neutriphil enzymes form lysosomes have after apoptosis? Do the proteolytic and hydrolytic enzymes result in further bone resorption?

Response 1: Recruitment and function of undisturbed neutrophilic PMNs are important in the inflammatory phase to initiate subsequent responses that lead to bone regeneration in fractures, and apparently, neutrophils would also play a relevant role in the early stages of osseointegration.

However, in the case of excessive post-traumatic inflammation, neutrophils may become overactivated or dysfunctional. Consequently, they secrete an altered cytokine profile, increase ROS production and undergo a massive NETosis, thus aggravating tissue damage and even damaging the surrounding healthy tissues. In fact, in eroded bones, the number of osteoclasts has been correlated with the abundance of infiltrated polymorphonuclear neutrophils (PMN). This is in relation to the clinical evidence that demonstrates the need to use atraumatic surgical protocols and the use of abundant irrigation to avoid overheating of the surgical area.

The interesting thing would be to demostrated if an implant with a contaminated surface could have the potential to generate overactivated or dysfunctional neutrophils, Since has been observed the activation of neutrophils  in the presence of nanoparticles. However, most studies evaluating neutrophil dysfunctions after trauma address its poor antimicrobial defense.

Point 2: Macrophage genotype impact, if known?

Response 2: I have no information about the impact of the macrophage genotype and osseointegration. But undoubtedly this is a topic that can become clinically relevant. There are genetic immunodeficiencies that affect the function of macrophages, what can contribute to chronic inflammatory diseases such as cancer, atherosclerosis and obesity. So it is definitely a field of research to develop, especially in relation to biomaterials.

Round  2

Reviewer 1 Report

The performed changes improved the quality of the manuscript. 

Author Response

Dear reviewer

Thanks for your response. I will looking for  someone with English as his/her mothers tongue, to make an English brush-up.

Kind regards

Luis